# Clinical utility of comprehensive genomic profiling in Japan: Result of PROFILE-F study

**Yasuko Aoyagi**[1,2]*, **Yoshihito Kano**[1,3], **Kohki Tohyama**[1], **Shotaro Matsudera**[1,4,5], **Yuichi Kumaki**[4], **Kenta Takahashi**[6], **Takahiro Mitsumura**[7], **Yohei Harada**[8], **Akemi Sato**[9], **Hideaki Nakamura**[9], **Eisaburo Sueoka**[9], **Naoko Aragane**[8], **Koichiro Kimura**[10], **Iichiro Onishi**[11], **Akira Takemoto**[12], **Keiichi Akahoshi**[13], **Hiroaki Ono**[13], **Toshiaki Ishikawa**[4], **Masanori Tokunaga**[2], **Tsuyoshi Nakagawa**[4], **Noriko Oshima**[6], **Reiko Nakamura**[6], **Masatoshi Takagi**[14], **Takahiro Asakage**[15], **Hiroyuki Uetake**[4], **Minoru Tanabe**[13], **Satoshi Miyake**[3], **Yusuke Kinugasa**[2], **Sadakatsu Ikeda**[1]*

1 Department of Precision Cancer Medicine, Center for Innovative Cancer Treatment, Tokyo Medical and Dental University, Tokyo, Japan, 2 Department of Gastrointestinal Surgery, Tokyo Medical and Dental University, Tokyo, Japan, 3 Department of Clinical Oncology, Tokyo Medical and Dental University, Tokyo, Japan, 4 Department of Specialized Surgeries, Tokyo Medical and Dental University, Tokyo, Japan, 5 First Department of Surgery, Dokkyo Medical University, Tochigi, Japan, 6 Department of Obstetrics and Gynecology, Tokyo Medical and Dental University, Tokyo, Japan, 7 Department of Respiratory Medicine, Tokyo Medical and Dental University, Tokyo, Japan, 8 Division of Hematology, Respiratory Medicine and Oncology, Faculty of Medicine, Saga University, Saga, Japan, 9 Department of Transfusion Medicine, Saga University Hospital, Saga, Japan, 10 Department of Radiology, Tokyo Medical and Dental University, Tokyo, Japan, 11 Department of Pathology, Tokyo Medical and Dental University, Tokyo, Japan, 12 Department of Bioresource Research Center, Tokyo Medical and Dental University, Tokyo, Japan, 13 Department of Hepato-Biliary-Pancreatic Surgery, Tokyo Medical and Dental University, Tokyo, Japan, 14 Department of Pediatrics, Tokyo Medical and Dental University, Tokyo, Japan, 15 Department of Head and Neck Surgery, Tokyo Medical and Dental University, Tokyo, Japan

* aoyagi.srg1@tmd.ac.jp (YA); ikeda.canc@tmd.ac.jp (SI)

## Abstract

### Introduction

Clinical sequencing has provided molecular and therapeutic insights into the field of clinical oncology. However, despite its significance, its clinical utility in Japanese patients remains unknown. Here, we examined the clinical utility of tissue-based clinical sequencing with FoundationOne® CDx and FoundationOne® Heme. Between August 2018 and August 2019, 130 Japanese pretreated patients with advanced solid tumors were tested with FoundationOne® CDx or FoundationOne® Heme.

### Results

The median age of 130 patients was 60.5 years (range: 3 to 84 years), and among them, 64 were males and 66 were females. Major cancer types were gastrointestinal cancer (23 cases) and hepatic, biliary, and pancreatic cancer (21 cases). A molecular tumor board had been completed on all 130 cases by October 31, 2019. The median number of gene alterations detected by Foundation testing, excluding variants of unknown significance (VUS) was 4 (ranged 0 to 21) per case. Of the 130 cases, one or more alterations were found in 123 cases (94.6%), and in 114 cases (87.7%), actionable alterations with candidates for therapeutic agents were found. In 29 (22.3%) of them, treatment corresponding to the gene

**Funding:** This study was supported by Chugai Pharmaceutical, Co. Ltd. Institutional funding number 2F142. The funders had no role in study design, data collection and analysis, decision to publish, or preparation of the manuscript.

**Competing interests:** The authors have declared that no competing interests exist.

alteration was performed. Regarding secondary findings, 13 cases (10%) had an alteration suspected of a hereditary tumor. Of the 13 cases, only one case received a definite diagnosis of hereditary tumor.

## Conclusions

Our study showed that clinical sequencing might be useful for detecting gene alterations in various cancer types and exploring treatment options. However, many issues still need to be improved.

## Introduction

In recent years, next-generation sequencing (NGS), which is a technology for analyzing the base sequence of DNA in a short time, has made it possible to examine a large amount of genomic information at once. Conventionally, cancer drug therapy has been selected based on organ of the primary tumor. However, it has been found that gene alterations in cancer cells can effect biology and treatment, even in cancers originating from the same organ. Therefore, cross-organ cancer genomic medicine is receiving a great deal of attention, in which cancer-related genes are comprehensively analyzed using a next-generation sequencer and a therapeutic drug is selected based on the results. Comprehensive examination of cancer-related genes from cancer tissues or blood samples is called comprehensive genomic profiling (CGP). The CGP tests, FoundationOne® CDx cancer genome profiling, and OncoGuide™ NCC Oncopanel System have been covered by Public health insurance since June 2019 and are widely adapted in Japan.

The results of examining the clinical utility of CGP tests have been reported in Japan as well. In Japan, CGP can only be used at the time of completion or expected completion of standard treatment to receive the reimbursement of Public Health Insurance. Therefore, it is difficult to make simple comparisons with foreign literature, and accumulation of data in Japan is necessary. In addition, FoundationOne® CDx cannot determine whether the detected mutations are of germline origin, but we may detect genetic alterations associated with hereditary tumors as secondary findings. Few reports of secondary findings in CGP tests are available in Japan. Therefore, to examine the clinical utility of the CGP test, we initiated an observational study using FoundationOne® CDx or FoundationOne® Heme and investigated the effect of the CGP test results on the treatment of patients. We also investigated secondary findings and outcomes in these tests.

## Methods

### Patients

We retrospectively reviewed 130 patients with advanced solid tumors who either progressed on, or were finishing standard systemic therapy. These patients underwent FoundationOne® CDx or FoundationOne® Heme between August 2018 and August 2019 under PROFILE-F study. The PROFILE-F study was approved by the institutional review board of Tokyo Medical and Dental University (TMDU; G2018-002) and registered in University Hospital Medical Information Network (UMIN; UMIN000028439).

### Sequencing and detection of genomic variances

Patients underwent clinical-grade CUA-approved next-generation sequencing that investigates the entire coding DNA sequencing of 324 genes with FoundationOne® CDx and DNA

sequencing of 406 genes and RNA sequencing of 265 genes with FoundationOne® Heme. Tumors were assessed for genomic aberrations, including insertions, deletion, base substitutions, copy number alterations, and fusions/ rearrangements. The methods for this type of comprehensive genomic profiling have been previously published, and extensive methods can be found elsewhere [1, 2].

## Definition of actionability

Actionable alteration is defined as a genomic alteration that satisfies the following conditions: 1) mechanistically, the gene is associated with cancer and has the data indicating therapeutic efficacy; and 2) a drug is available for human use either in an antibody or a small molecule compound with low IC 50 concentration [3, 4].

## Molecular tumor board

After genomic test results, each case was discussed at the molecular tumor board (MTB) with specialists, such as medical oncologists, pathologists, radiologists, bioinformaticians, genetic counselors, clinical research coordinators, and treating physicians. These members deliberated actionable genomic alterations and treatment options based on the patient's medical history, treatment history, family history, imaging findings, histopathological findings, and genetic test results.

## Results

### Patients and characteristics

Between August 2018 and August 2019, 130 patients with advanced solid tumors who progressed with or were finishing standard systemic therapy or with rare cancers participated in the PROFILE-F study and underwent FoundationOne® CDx or FoundationOne® Heme. The median age of 130 patients at specimen exam date was 60.5 years (range: 3 to 84 years), and among them, 64 were males and 66 were females (Table 1). There were 16 patients (12.3%) with no previous chemotherapy by the date of test submission, 27 patients (20.8%) with one line and 87 patients (66.7%) with two or more lines. A total of 28 diverse cancer types were observed in 130 patients (Table 1). All patient populations were Japanese. The breakdown of FoundationOne® CDx and FoundationOne® Heme is shown in Table 2.

### The common alterations and differences by each cancer type

The most commonly altered genes excluding VUS in all cancer types are shown in Fig 1A. Only alterations seen in at least 5% of patients were shown. The three most frequent alterations observed were *TP53* (n = 72, 55.4%), *CDKN2A* (n = 29, 22.3%), *KRAS* (24, n = 18.5%). As shown in Fig 1B–1E, the most frequent gene alterations observed varied by tumor types.

### Alterations and actionability

Table 3 shows the alteration and actionability in 130 patients with each cancer type, while Fig 2 displays the flowchart of actionability of 130 patients. Overall, 123 (94.6%) of 130 patients had detectable alteration(s). Of the 130 patients, 114 (87.7%) had at least one actionable alteration. The median number of alterations (except VUS) per patient of all cancer types was 4 (range 0–21), and the median number of actionable alterations was 3 (range 0–12). The cancer types with the highest median number of actionable alterations were NSCLC (non-small cell lung cancer) (median: 5.5; range 1–8), esophagus caner (median: 5; range 2–12), ovary cancer

**Table 1. Patients characteristics.**

| Characteristics | No. of patients (%) | |
|---|---|---|
| Age at specimen exam date, years | | |
| Median(range) | 60.5 | (3–84) |
| Gender | | |
| Male | 64 | (49.2%) |
| Female | 66 | (50.8%) |
| Line of previous chemotherapy | | |
| 0 | 16 | (12.3%) |
| 1 | 27 | (20.8%) |
| ≥2 | 87 | (66.9%) |
| Type of cancer | | |
| Neuroendocrine tumor | 14 | (10.8%) |
| Pancreas cancer | 12 | (9.2%) |
| Breast cancer | 12 | (9.2%) |
| Colorectal cancer | 11 | (8.5%) |
| Head and Neck cancer | 8 | (6.2%) |
| Sarcoma | 8 | (6.2%) |
| Esophagus cancer | 7 | (5.4%) |
| CUP | 7 | (5.4%) |
| Biliary cancer | 6 | (4.6%) |
| NSCLC | 6 | (4.6%) |
| Ovary cancer | 5 | (3.8%) |
| Uterus cancer | 5 | (3.8%) |
| Urologic cancer | 3 | (2.3%) |
| Stomach cancer | 2 | (1.5%) |
| Liver cancer | 2 | (1.5%) |
| SCLC | 2 | (1.5%) |
| Other * | 20 | (15.4%) |

Abbreviations: CUP, Cancer of unknown primary origin; NSCLC, Non-Small Cell Lung Cancer; SCLC, Small Cell Lung Cancer.

* Other cancers include hemangiopericytoma, chordoma, adrenal carcinoma, thymic carcinoma, peripheral schwannoma, peritoneal mesothelioma, nephroblastoma, neuroblastoma, extramammary Paget's disease, chondrosarcoma, allantoic carcinoma, and primary intraosseous carcinoma.

(median: 5; range 1–5), and CUP (cancer of unknown primary origin) (median: 5; range 2–10). Meanwhile, the cancer type with the lowest median number of actionable alterations was neuroendocrine tumor (median: 0; range 0–6). We added Fig 3 showing evidence level defined by C-CAT (The Center for Cancer Genomics and Advanced Therapeutics) [5]. Of 130 patients, 29 (22.3%) received the treatment corresponding to the gene alteration. The cancer type with the highest rate of patients who received the treatment corresponding to the gene alteration was SCLC (small cell lung cancer) (1 in 2 or 50.0%). Table 4 shows the extended information in 130 patients with each cancer type. We also obtained the results of tumor alteration burden (TMB) and microsatellite status (MS). The median number of TMB was 4 (range 0–34). The cancer type with the highest median number of TMB was CUP (median:14; range 3–29). Only 1 patient who was diagnosed with cancer of unknown primary origin reaped the result of MS-high.

**Table 2. Types of tests.**

| Types of tests | No. of specimens submitted (%) |
|---|---|
| Foundation One® CDx | 139(87.4%) |
| Foundation One® Heme | 20(12.6%) |
| all specimens | 159(100.0%) * |

*There is a discrepancy between the number of specimens and the number of patients, including patients who submitted several tests for failure reports, patients who submitted both FoundationOne® CDx and FoundationOne® Heme, and patients who submitted specimens from multiple sites (primary and metastatic).

## Treatment corresponding to the gene alteration

Of 130 patients, 29 (22.3%) were treated based on comprehensive genomic profiling. We divided the patients according to the type of therapeutic drug used (Table 5). The three categories used are approved drug (Approved), investigational drug (Investigational), and off-label use (Off-label). Among the treatments that patients received, approved drug was 51.7% (15 out of 29), investigational drug was 31.0% (9 out of 29), and off-label use was 17.2% (5 out of 29). Among patients who received approved treatment, 5 patients received immune checkpoint inhibitors, nivolumab, or pembrolizumab (Nos. 9, 14, 15, 24, and 28), and 5 patients received PARP inhibitors or platinum-based anticancer agents for homologous recombination repair-related gene alterations (Nos. 10, 12, 13, 18, and 29). The best responses in patients who received approved treatment were CR 6.7%, PR 33.3%, SD 13.3%, PD 26.7%, and N.D. 20.0%. Among patients who received investigational drugs, 3 took a combination of pertuzumab and trastuzumab for *ERBB2* amplification (Nos. 23, 26, and 27) and 1 received olaparib for *ATM* alteration (No. 25). The administration of these investigational drugs were conducted at our own facility. Since other investigational drugs were conducted at other institutions, the patients were referred to them accordingly. No. 20 patient was a case with *EGFR* uncommon alterations which were not detected by the initial PNA LNA PCR-Clamp method, but were able to be detected by FoudantionOne® CDx [6]. Although not included in Table 5, there was a case in which NGS influenced the treatment strategy. The patient had been treated for pathologically diagnosed primary intrahepatic cholangiocarcinoma before NGS. But She had a history of pancreatic cancer surgery, and the NGS results of the pancreatic resection specimen and liver resection specimen matched, so the MTB discussion changed the diagnosis to liver metastasis of pancreatic cancer. She responded to a pancreatic cancer regimen.

## Secondary findings

In this study, we detected somatic alterations in tumor tissue to explore cancer treatment options. However, as a result of discussions at the molecular tumor board based on factors like detected alterations, family history, and age of onset, there were 13 patients (10%) suspected of having a hereditary tumor as secondary findings (Fig 4). Of the 13 patients, 4 (3.1%) underwent a test for a definite diagnosis of hereditary tumor. Only 1 patient (0.8%) reached a definite diagnosis for it. Fig 4 also shows the reason patients suspected of having a hereditary tumor were not tested for definite diagnosis for hereditary tumor. Four patients had not yet received genetic counseling, 3 died before genetic counseling, 1 disagreed to know suspected hereditary tumors, and 1 refused a definitive test after genetic counseling.

## Discussion

Recently, tumor agonistic genomic medicine, which uses somatic or germline genetic alterations to guide decisions about treatment choice, has been attracting attention. Many

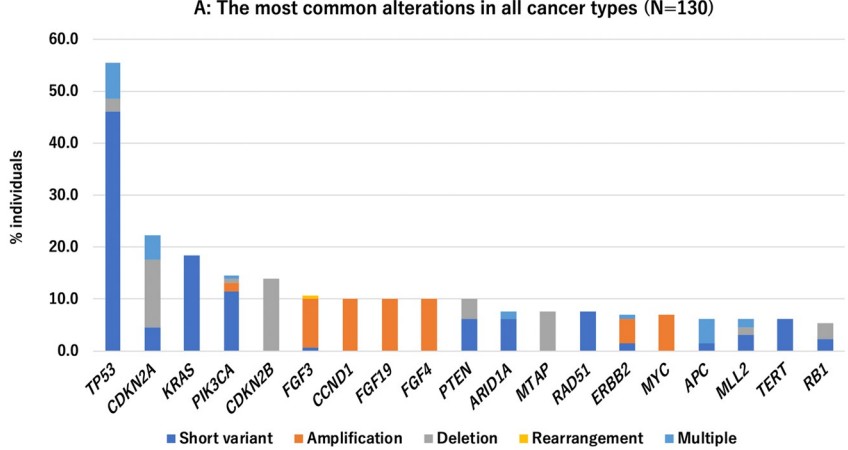

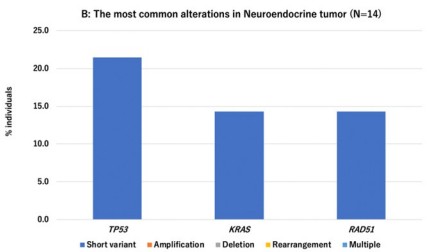

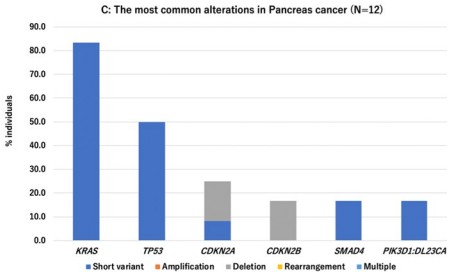

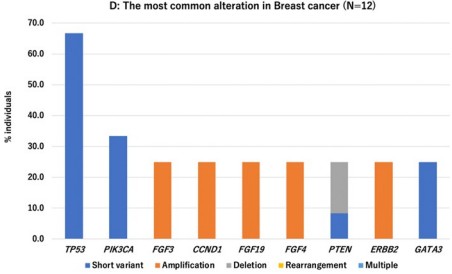

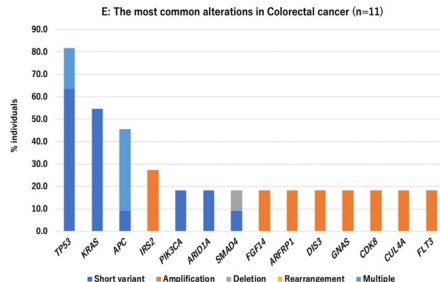

**Fig 1. The most common alterations in all cancer types (N = 130). A.** The most commonly altered genes in all cancer types, seen in at least 5% of patients, were shown. The three most frequent alterations observed were *TP53* (n = 72, 55.4%), *CDKN2A* (n = 29, 22.3%) and *KRAS* (24, n = 18.5%). **B.** The most common alterations in Neuroendocrine tumor (N = 14). **C.** The most common alterations in Pancreas cancer (N = 12). **D.** The most common alterations in Breast cancer (N = 12). **E.** The most common alterations in Colorectal cancer (N = 11).

institutions are investigating the clinical utility of the Comprehensive Genomic Profiling (CGP) test, which is indispensable in cancer genomic medicine. One of the indicators of clinical utility of the CGP test is whether patients actually received the treatment recommended by the CGP test. In the present study, we investigated the impact of the CGP tests, FoundationOne® CDx or FoundationOne® Heme, on patients' treatment choices.

The clinical utility of CGP testing has been reported previously in Europe and the U.S. Sohal DPS et al. reported that 11% (24 of 223) of patients received the recommended treatment

**Table 3. Alterations and actionability in 130 patients with each cancer types.**

| Cancer types | No. of individuals | No. of individuals with detectable alteration(s) (%) | No. of individuals with ≥1 alterations (%) | Median no. of alterations (range) | No. of individuals with ≥1 actionable alterations (%) | Median no. of actionable alteration(s) (range) | No. of individuals who received molecular-targeted therapy (%) |
|---|---|---|---|---|---|---|---|
| Neuroendocrine tumor | 14 | 14(100%) | 8(57.1%) | 1(0–8) | 7(50.0%) | 0(0–6) | 1(7.1%) |
| Pancreas cancer | 12 | 12(100%) | 12(100%) | 3.5(2–9) | 8(66.7%) | 2.5(1–6) | 3(25.0%) |
| Breast cancer | 12 | 12(100%) | 12(100%) | 5(1–14) | 12(100%) | 4(1–7) | 3(25.0%) |
| Colorectal cancer | 11 | 10(90.9%) | 10(90.9%) | 6.5(3–16) | 10(90.9%) | 4.5(2–8) | 4(36.4%) |
| Head and Neck cancer | 8 | 8(100%) | 8(100%) | 4.5(1–11) | 8(100%) | 4(1–6) | 2(25.0%) |
| Sarcoma | 8 | 8(100%) | 8(100%) | 5(2–9) | 8(100%) | 2.5(2–6) | 2(25.0%) |
| Esophagus cancer | 7 | 7(100%) | 7(100%) | 6(2–21) | 7(100%) | 5(2–12) | 1(14.3%) |
| CUP | 7 | 7(100%) | 7(100%) | 8(3–13) | 7(100%) | 5(2–10) | 3(42.9%) |
| Biliary cancer | 6 | 6(100%) | 6(100%) | 2.5(1–6) | 6(100%) | 2(1–4) | 0(0%) |
| NSCLC | 6 | 4(66.7%) | 4(66.7%) | 6.5(2–10) | 4(66.7%) | 5.5(1–8) | 1(16.7%) |
| Ovary cancer | 5 | 5(100%) | 5(100%) | 6(1–8) | 5(100%) | 5(1–5) | 1(20.0%) |
| Uterus cancer | 5 | 5(100%) | 5(100%) | 2(1–7) | 5(100%) | 1(1–6) | 2(40.0%) |
| Urologic cancer | 3 | 3(100%) | 3(100%) | 2(2–8) | 3(100%) | 2(1–8) | 0(0%) |
| Stomach cancer | 2 | 2(100%) | 2(100%) | 5(3–7) | 2(100%) | 2(2) | 0(0%) |
| Liver cancer | 2 | 2(100%) | 2(100%) | 4.5(4–5) | 2(100%) | 4(4) | 0(0%) |
| SCLC | 2 | 1(50%) | 1(50.0%) | 3(3) | 1(50.0%) | 2(2) | 1(50.0%) |
| Other | 20 | 17(85%) | 17(85.0%) | 2(1–11) | 15(75.0%) | 2(0–9) | 5(25.0%) |
| All | 130 | 123(94.6%) | 117(90.0%) | 4(0–21) | 114(87.7%) | 3(0–12) | 29(22.3%) |

*without VUS (variants of unknown significance).

based on the CGP test using the FoundationOne platform [8]. Hirshfield KM et al. reported that the rate of clinical intervention based on the CGP test using the FoundationOne platform was 35% (31 of 92), including genetically guided therapy, diagnostic modification, and trigger for germline genetic testing [9]. In the NCI-MATCH trial, a clinical trial in which subjects' tumor samples are screened with a CGP test and subjects with potentially targetable genetic alterations are entered into a clinical trial corresponding to that genetic alteration, the percentage of patients assigned to treatment was 17.8% [10].

In Japan, CGP can only be used at the time of completion or expected completion of standard treatment to receive the reimbursement of Public Health Insurance [5]. Therefore, it is difficult to make simple comparisons with foreign literature, and accumulation of data in Japan is necessary. In Japan, OncoGuide™ NCC Oncopanel System and FoundationOne®️ CDx cancer genome profiling are currently reimbursed by Public Health Insurance. Other research-based NGS (e.g., PleSSision-160, CANCERPLEX, OncoPrime) results have been reported in Japan [11–14]. Hayashi et al. reported that of 20 pancreatic cancer patients who underwent a targeted amplicon exome sequencing for 160 cancer-related genes (PleSSision-160), 100%(20/20) had actionable gene alterations, 35%(7/20) had druggable alterations detected, and only 10%(2/20) could be treated with therapeutic agents based on the results of genomic testing [11]. Saotome et al. reported that among the ovarian tumor patients who underwent PleSSision-160, actionable alterations were detected in 90.9%(80/88) and druggable alterations were detected in 40.9%(36/88) [12]. Kou et al. reported that among 85 patients with cancers of unknown primary site, rare tumors, or any solid tumors that were refractory to standard chemotherapy who underwent an NGS-based multiplex gene assay (OncoPrime), 69

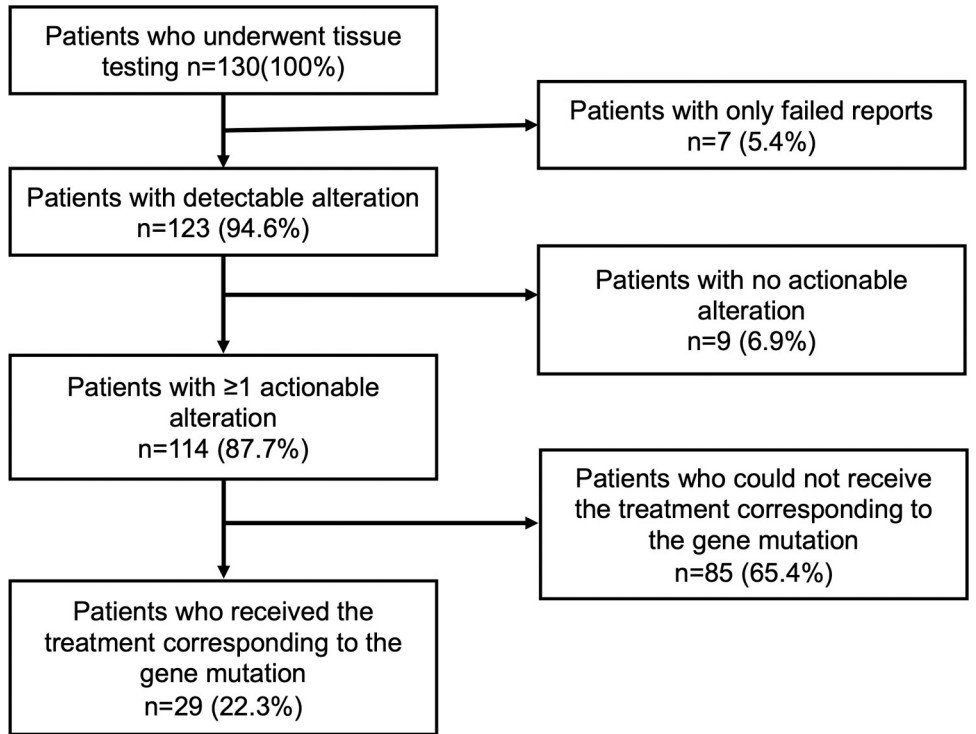

**Fig 2. Flowchart of actionability of 130 patients.** 123 (94.6%) of 130 patients had detectable alteration(s). There were 114 (87.7%) patients with at least one actionable alteration. 29 (22.3%) patients received the treatment corresponding to the gene alteration.

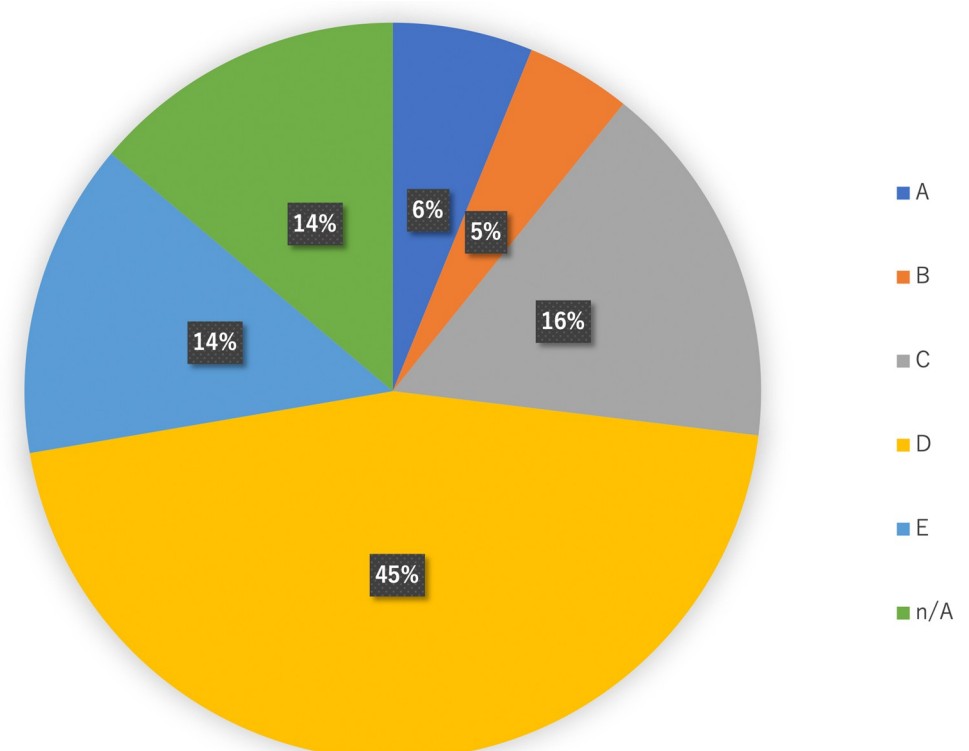

**Fig 3. Levels of evidence defined by C-CAT.** Alterations with evidence level D or higher were detected in 94 (72.3%) of 130 patients.

**Table 4. Extended information in 130 patients with each cancer types.**

| Cancer types | No. of individuals | Median TMB* (range) | No. of individuals with TMB ≥10 (%) | No. of individuals with MS**-High (%) | No. of individuals with ≥1 clinical trial options (%) |
|---|---|---|---|---|---|
| Neuroendocrine tumor | 14 | 1(0–6) | 0(0%) | 0(0%) | 6(42.9%) |
| Pancreas cancer | 12 | 3(0–6) | 0(0%) | 0(0%) | 8(66.7%) |
| Breast cancer | 12 | 2.5(0–9) | 0(0%) | 0(0%) | 11(91.7%) |
| Colorectal cancer | 11 | 3.5(0–11) | 1(9.1%) | 0(0%) | 10(90.9%) |
| Head and Neck cancer | 8 | 5.5(1–34) | 1(12.5%) | 0(0%) | 8(100%) |
| Sarcoma | 8 | 2.5(1–13) | 2(25.0%) | 0(0%) | 6(75.0%) |
| Esophagus cancer | 7 | 4(3–14) | 1(14.3%) | 0(0%) | 6(85.7%) |
| CUP | 7 | 14(3–29) | 4(57.1%) | 1(14.3%) | 5(71.4%) |
| Biliary cancer | 6 | 4(3–9) | 0(0%) | 0(0%) | 4(66.7%) |
| NSCLC | 6 | 2.5(0–10) | 1(16.7%) | 0(0%) | 4(66.7%) |
| Ovary cancer | 5 | 4(0–14) | 1(20.0%) | 0(0%) | 5(100%) |
| Uterus cancer | 5 | 3(3–8) | 0(0%) | 0(0%) | 4(80.0%) |
| Urologic cancer | 3 | 3(0–23) | 1(33.3%) | 0(0%) | 3(100%) |
| Stomach cancer | 2 | 5(5) | 0(0%) | 0(0%) | 1(50.0%) |
| Liver cancer | 2 | 4.5(4–5) | 0(0%) | 0(0%) | 2(100%) |
| SCLC | 2 | 9(9) | 0(0%) | 0(0%) | 0(0%) |
| Other | 20 | 3.5(0–9) | 0(0%) | 0(0%) | 10(50.0%) |
| All | 130 | 4(0–34) | 12(9.2%) | 1(0.8%) | 97(74.6%) |

*Tumor mutation burden: without cannot determined patients.

**Microsatellite status.

patients had potentially actionable alterations detected. 9(13.0%) of 69 patients received a subsequent therapy based on the NGS assay results [14]. According to a report by Sunami et al. using the reimbursed OncoGuide™ NCC Oncopanel System, of 230 patients with advanced solid tumors, 111 (59.4%) of which harbored actionable gene alterations and twenty-five (13.3%) cases have since received molecular-targeted therapy according to their gene alterations [15]. Takeda et al reported that among the 175 patients who underwent FoundationOne® CDx, 174 had at least one known or likely pathogenic gene alteration, and 24 of these patients (14%) received corresponding targeted therapy [16]. In our study, 22.3% of patients received treatment based on CGP results, which is almost the same as other Japanese reports. We defined actionable alteration as a genomic alteration that satisfies the following conditions: 1) mechanistically, the gene is associated with cancer and has the data indicating therapeutic efficacy; and 2) a drug is available for human use either in an antibody or a small molecule compound with low IC 50 concentration [3, 4]. But this is an area of on-going debate. It is difficult to compare the number of actionable alterations with other Japanese reports. So we added Fig 3 showing evidence level defined by C-CAT (The Center for Cancer Genomics and Advanced Therapeutics) [5]. Alterations with evidence level D or higher were detected in 94 (72.3%) of 130 patients. The reasons for not receiving molecular targeted therapy despite the detection of actionable mutations were as follows: 1) investigational drugs were only available overseas or in distant parts of Japan, 2) clinical trial recruitment had already ended, 3) comorbidity or poor PS prevented participation in the trial, or 4) the disease had progressed to the point where treatment was not indicated. 1) and 2) suggests that there are inter-facility and inter-regional disparities in access to investigational drugs. Correcting the disparities in

**Table 5. Patients who received the treatment corresponding to the gene alteration.**

| No. | diagnosis | Age (years) | Gender | Lines of previous CTx | Targeted gene aberration | Drug | Category |
|---|---|---|---|---|---|---|---|
| 1 | Breast cancer | 60 | F | 6 | CCND1 amplification | Palbociclib | Approved |
| 2 | Breast cancer | 43 | F | 11 | PIK3CA N345K, AKT1 amplification | Everolimus | Approved |
| 3 | Breast cancer | 43 | F | 6 | FGFR1 amplification | Combination of TAS-117 and TAS120 | Investigational |
| 4 | Colorectal cancer | 65 | M | 7 | APC R232* | Wnt inhibitor | Investigational |
| 5 | Colorectal cancer | 70 | F | 5 | PIK3CA E545K | mTOR inhibitor | Investigational |
| 6 | Colorectal cancer | 71 | F | 4 | FLT3 amplification | Regorafenib | Approved |
| 7 | Small intestinal cancer | 66 | M | 2 | APC E1379*, APC K534*, APC splice site 835-8A>G | β-catenin inhibitor | Investigational |
| 8 | Esophageal cancer | 64 | M | 0 | TMB high | Pembrolizumab | Off-label |
| 9 | Sarcoma of the esophagus | 63 | F | 0 | TMB high | Nivolumab | Approved |
| 10 | Uterine sarcoma | 58 | F | 3 | RAD51B loss | IP(Ifomide, CDDP and Mesna) | Approved |
| 11 | Cervical cancer | 55 | F | 4 | ARID1A E1647* | ATR inhibitor | Investigational |
| 12 | Cervical cancer | 79 | F | 1 | BRCA1 S153fs*5 | 1) CBDCA, 2) Olaparib | Approved |
| 13 | Ovarian cancer | 72 | F | 4 | BRCA2 R2318* | CDDP | Approved |
| 14 | Tongue cancer | 63 | M | 2 | PD-L1 TPS70%† | Nivolumab | Approved |
| 15 | Maxillary cancer | 50 | M | 0 | TMB high | Nivolumab | Approved |
| 16 | Pancreatic cancer | 64 | M | 3 | PIK3CA H1047R | Copanlicib | Off-label |
| 17 | Pancreatic cancer | 43 | M | 3 | KRAS G12D | Combination of Trametinib and Hydroxychlorquine | Off-label |
| 18 | Pancreatic cancer | 72 | F | 3 | ATM R2993* | FOLFOX | Approved |
| 19 | Small-cell lung cancer | 63 | M | 1 | TMB 9 muts/Mb | Nivolumab | Off-lavel |
| 20 | Lung adenocarcinoma | 70 | F | 12 | EGFR G719D, EGFR E709A§ | Afatinib | Approved |
| 21 | Duodenal neuroendocrine tumor | 58 | F | 6 | BRCA1 rearrangement | Olaparib | Off-label |
| 22 | Hemangiopericytoma | 56 | F | 0 | NAB2-STAT6 fusion | Pazopanib | Approved |
| 23 | Urachal cancer | 40 | M | 1 | ERBB2 amplification | Combination of Pertuzumab and Trastuzumab | Investigational |
| 24 | Malignant peripheral nerve sheath tumor | 63 | F | 1 | TMB 9 muts/Mb, MSH6 N1307fs*9 | Pembrolizumab | Approved |
| 25 | Nephroblastoma | 9 | M | 4 | ATM K2749I(VUS)‖ | Olaparib | Investigational |
| 26 | Extramammary Paget's disease | 55 | M | 1 | ERBB2 amplification | Combination of Pertuzumab and Trastuzumab | Investigational |
| 27 | Cancer of unknown primary origin | 70 | F | 1 | ERBB2 amplification | Combination of Pertuzumab and Trastuzumab | Investigational |
| 28 | Cancer of unknown primary origin | 58 | M | 1 | MSI high, TMB high | Pembrolizumab | Approved |
| 29 | Cancer of unknown primary origin | 72 | F | 1 | RAD51D K91fs*13 | Olaparib | Approved |

TMB more than 10 muts/Mb is defined as TMB-high.

†: PD-L1 TPS was measured as an optional service of FoundationOne® CDx.

§: A case with EGFR uncommon alterations which were not detected by the initial PNA LNA PCR-Clamp method, but were able to be detected by FoudantionOne® CDx. [6].

‖: Although reported as VUS in the FoundationOne® Heme report, a preclinical study has shown the sensitivity of PARP inhibitor to this alteration (ATM p.K2749I) [7].

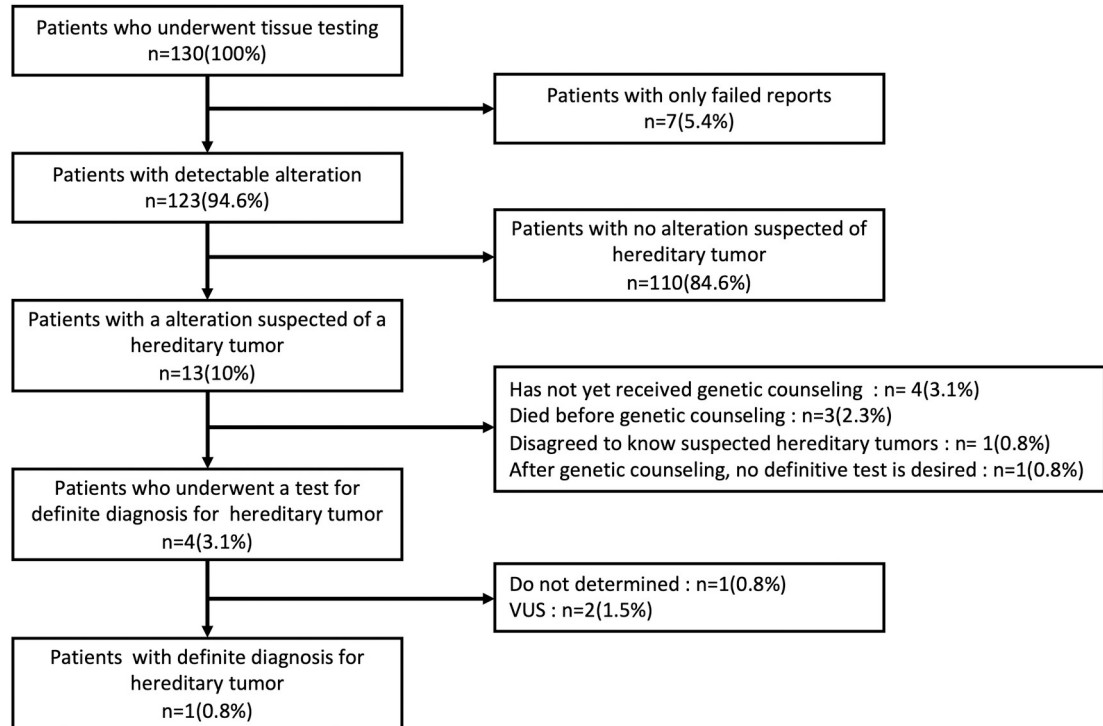

**Fig 4. Outcome of patients with suspected secondary findings.** Of the 130 patients, 13 (10%) were suspected of having a hereditary tumor. Only 1 patient (0.8%) reached a definite diagnosis for it.

obtaining information and referrals for investigational drugs is desirable. Cases such as 3) and 4) may be influenced by the fact that CGP in Japan Public Health Insurance can be used only at the time of completion or prospective completion of standard treatment. If the CGP could be performed from the start of cancer treatment, it might lead to better treatment choices.

We experienced a patient whose diagnosis was changed in the MTB (called expert panel in Japan) discussion due to the CGP results and who was treated based on the new diagnosis. The clinical usefulness of the CGP test may be enhanced by a comprehensive discussion of the optimal treatment for the patient based on the CGP test results in the MTB. In Japan, an expert panel is required as a condition of insurance treatment, and this is expected to boost the usefulness of the CGP test. However, expert panels in Japan differ from facility to facility, and standardization is an issue to be addressed in the future [17].

Hirshfield et al. reported that tumor sequencing results can be a trigger for germline testing [9]. In a previous report, hereditary tumors were suspected in five (6.2%) of 80 patients, three of whom underwent definitive testing for hereditary tumors, and two (2.5%) of whom were confirmed [14]. In the present study, we experienced a case in which the results of CGP testing to explore treatment options led to the diagnosis of hereditary tumors. However, although hereditary tumors were suspected in 13 (10%) out of 130 patients, only 4 (3.1%) patients underwent tests for confirmatory germline testing. This may be since hereditary tumors are still not widely recognized in Japan, or the priority to explore hereditary possibility was lower than pursuing the treatment for the patient. The diagnosis of hereditary tumors can not only lead to the treatment of the patient, but also to the prevention and early detection of cancer in the next generation. Taking all these into consideration, spreading awareness and accurate knowledge about hereditary tumors should be considered as important aspects of cancer treatment.

There are several limitations in our study. First, it was difficult to make statistical comparisons with the foreign literature due to the nature of the Public Health insurance system in Japan. Second, the definition of actionable alteration is still controversial, and we were unable to compare the actionable alteration rate with other Japanese literature. Third, because this was a retrospective study, we were not able to follow the progress of some patients who had been treated at other hospitals.

## Conclusion

This study showed that CGP tests might be useful for detecting gene alterations in various cancer types and exploring treatment options. However, many issues still require improvement, including better access to investigational and off-label use drugs, standardization of MTB, and understanding of hereditary tumors.

## Supporting information

**S1 Table. All Patients data.** This table contains all patient characteristics and genetic information.
(XLSX)

## Acknowledgments

We thank Ms. Junko Yokobori, Ms. Mika Ohki, and Ms. Yumi Kobayashi for their exceptional support of this study as clinical research coordinator (CRC). Ms. Eriko Takamine, a certified genetic counselor, helped to organize molecular tumor boards and genetic counseling of patients who had potential germline mutations. We also would like to extend our gratitude to Ms. Tomomi Urata and Ms. Akiko Noguchi for their administrative support.

## Author Contributions

**Conceptualization:** Sadakatsu Ikeda.

**Data curation:** Yasuko Aoyagi, Yohei Harada, Sadakatsu Ikeda.

**Formal analysis:** Yasuko Aoyagi.

**Funding acquisition:** Sadakatsu Ikeda.

**Investigation:** Yasuko Aoyagi, Yoshihito Kano, Kohki Tohyama, Shotaro Matsudera, Yuichi Kumaki, Kenta Takahashi, Takahiro Mitsumura, Yohei Harada, Akemi Sato, Iichiro Onishi, Akira Takemoto, Keiichi Akahoshi, Hiroaki Ono, Toshiaki Ishikawa, Masanori Tokunaga, Tsuyoshi Nakagawa, Noriko Oshima, Reiko Nakamura, Masatoshi Takagi, Takahiro Asakage, Hiroyuki Uetake, Minoru Tanabe, Satoshi Miyake, Yusuke Kinugasa, Sadakatsu Ikeda.

**Methodology:** Sadakatsu Ikeda.

**Project administration:** Yasuko Aoyagi, Yoshihito Kano, Sadakatsu Ikeda.

**Supervision:** Yoshihito Kano, Hideaki Nakamura, Eisaburo Sueoka, Naoko Aragane, Koichiro Kimura, Satoshi Miyake, Yusuke Kinugasa, Sadakatsu Ikeda.

**Visualization:** Yasuko Aoyagi.

**Writing – original draft:** Yasuko Aoyagi.

**Writing – review & editing:** Yasuko Aoyagi, Yoshihito Kano, Sadakatsu Ikeda.

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
