## [Decision Letter · Decision Letter 0]

3 Dec 2021

PONE-D-21-35160Clinical utility of Comprehensive Genomic Profiling in Japan: Result of PROFILE-F study.PLOS ONE

Dear Dr. Aoyagi,

Thank you for submitting your manuscript to PLOS ONE. After careful consideration, we feel that it has merit but does not fully meet PLOS ONE’s publication criteria as it currently stands. Therefore, we invite you to submit a revised version of the manuscript that addresses the points raised during the review process.Please ensure that your decision is justified on PLOS ONE’s publication criteria and not, for example, on novelty or perceived impact.

We look forward to receiving your revised manuscript.

Kind regards,

Hyunseok Kang, MD, MPH

Academic Editor

PLOS ONE

Journal Requirements:

Reviewers' comments:

Reviewer's Responses to Questions

**Comments to the Author**

1. Is the manuscript technically sound, and do the data support the conclusions?

Reviewer #1: Partly

Reviewer #2: Partly

2. Has the statistical analysis been performed appropriately and rigorously? 

Reviewer #1: N/A

Reviewer #2: No

3. Have the authors made all data underlying the findings in their manuscript fully available?

Reviewer #1: No

Reviewer #2: Yes

4. Is the manuscript presented in an intelligible fashion and written in standard English?

Reviewer #1: No

Reviewer #2: Yes

5. Review Comments to the Author

Reviewer #1: Comments to the Author

The authors reported the clinical utility of CGP test in Japan using commercial targeted sequencing system, and they concluded it might be useful for detecting gene alterations in various cancer types and exploring treatment options.

Major comments:

1. The definition of actionability in this report is ambiguous. Please redefine the actionability using evidence levels in accordance with previously reported guidance.

2. Please mention the status of genomic medicine in Japan more exactly. For example, detection rate of actionable genomic alteration and implementation rate of genotype-matched treatment in other platforms such as OncoPrime, P5, PleSSision, CANCERPLEX, CLHURC and so on.

3. VUS should not be contained in Fig1A-G.

4. Please mention the limitation of this study.

5. The authors concluded that many issues still need to be improved in conclusion of the abstract. However, those many issues are not referred in this paper. Please discuss the problems of genomic medicine in Japan more.

6. Please attach all genomic information data of the patients in Supplement files.

Minor comments:

1. Please classify ‘Gastrointestinal cancer’, ‘Hepatic Biliary and Pancreatic cancer’, ‘Lung Cancer,’ ‘Gynecologic cancer’ into esophagus, stomach, colorectal, liver, biliary, pancreas, NSCLC, SCLC, ovary, uterus.

2. Patient No.19 should be removed from the lists of patients who received the treatment corresponding to the gene alteration in Table 3. This is not the case of genotype-matched treatment.

3. What is the definition of TMB-High in this report?

Reviewer #2: Overall, this study’s goal is to describe the utility of Clinical genomic sequencing in Japan, a country that historically puts significant limitations on who can receive NGS sequencing. As a consequence, limited data is available from this country regarding NGS clinical utility.

I think the data they have collected could be useful to have in the literature, but as of now, the authors present the data purely in an observational format without providing any context or statistical comparisons about how their data in Japanese populations compare to previously published Foundation CDx retrospective studies.

Given the only novelty of this paper would be how numbers of clinically relevant alterations compare in populations in Japan vs other populations, statistical comparisons between this study's population and others are important to include as part of the results in order to provide context.

I’m unsure about the use of the word “Actionability” in the context of their document. I believe the right phrase may instead be “Clinical Utility”. In my thought process, “actionability” would imply an alteration that would change the treatment options available to a patient. If possible, It would be helpful to report which of the alterations reported in Foundation Dx fall under “actionable”. Based on the frequency reported (including an average of 3 “actionable” alterations/patient) I have to assume many of these alterations are only clinically significant in their prognostic abilities, not in their ability to open therapeutic options. Examples of these are TP53 or KRAS, which are clinically useful from a prognostic standpoint, but I feel calling them “actionable” may be disingenuous. However, I do not feel strongly about this semantic distinction if it doesn't bother the editor or other reviewers.

From a recommendation standpoint, if they are able to provide statistical context for how the results in their population compare with previous Foundation CDx reports in USA and Europe, including alteration prevalence and power to detect statistical differences, I think the paper is worth adding to the literature via PLOSOne publication.

(more detailed comments in the attached document).

6. PLOS authors have the option to publish the peer review history of their article (what does this mean?). If published, this will include your full peer review and any attached files.

Reviewer #1: **Yes: **Hideyuki Hayashi

Reviewer #2: **Yes: **Travis Zack

---

## [Author Response · Author response to Decision Letter 0]

27 Feb 2022

Journal Requirements:

1. Please ensure that your manuscript meets PLOS ONE's style requirements, including those for file naming. The PLOS ONE style templates can be found at  https://journals.plos.org/plosone/s/file?id=wjVg/PLOSOne_formatting_sample_main_body.pdf and  https://journals.plos.org/plosone/s/file?id=ba62/PLOSOne_formatting_sample_title_authors_affiliations.pdf

Reply: I have checked and corrected the above matters.

Reply: I confirm and correct the above statement.

Reply: I upload our minimal underlying data set as Supporting Information files.

Reply: I correct the above matter.

Reviewer #1

The authors reported the clinical utility of CGP test in Japan using commercial targeted sequencing system, and they concluded it might be useful for detecting gene alterations in various cancer types and exploring treatment options.

Reply: We thank the Reviewer 1’s favorable comments.

Major comments:

1. The definition of actionability in this report is ambiguous. Please redefine the actionability using evidence levels in accordance with previously reported guidance.

Reply: We thank reviewer’s comment regarding actionability. This is an area of on-going debate. We added Fig 3 showing evidence level defined by C-CAT (The Center for Cancer Genomics and Advanced Therapeutics) and elaborated in discussion. 

2. Please mention the status of genomic medicine in Japan more exactly. For example, detection rate of actionable genomic alteration and implementation rate of genotype-matched treatment in other platforms such as OncoPrime, P5, PleSSision, CANCERPLEX, CLHURC and so on.

Reply: We summarized recent reports of clinical utility of CGP in Japan and commented in discussion. 

3. VUS should not be contained in Fig1A-G.

Reply: We revised the figure excluding VUSs. 

4. Please mention the limitation of this study.

Reply: We thank the reviewer to remind us of including the limitation of the study. We added limitation in discussion.

5. The authors concluded that many issues still need to be improved in conclusion of the abstract. However, those many issues are not referred in this paper. Please discuss the problems of genomic medicine in Japan more.

Reply: We thank this reviewer for his remarks. We added the problems of genomic medicine in Japan in discussion.

6. Please attach all genomic information data of the patients in Supplement files.

Reply: We added Supporting Information files that includes key genomic data used in this study. 

Minor comments:

1. Please classify ‘Gastrointestinal cancer’, ‘Hepatic Biliary and Pancreatic cancer’, ‘Lung Cancer,’ ‘Gynecologic cancer’ into esophagus, stomach, colorectal, liver, biliary, pancreas, NSCLC, SCLC, ovary, uterus.

Reply: Thank you for your suggestion. We revised figures with more detailed classifications.

2. Patient No.19 should be removed from the lists of patients who received the treatment corresponding to the gene alteration in Table 3. This is not the case of genotype-matched treatment.

Reply: Thank you for your comment. We removed the patient No. 19 from the table 3 and revised result and discussion sections. 

3. What is the definition of TMB-High in this report?

Reply: TMB more than 10 muts/Mb is defined as TMB-high. In case No. 19, the patient had TMB 9 muts/Mb (intermediate), but since there were no other genetic alterations to target for treatment, after discussion among the experts in the field, it was decided to target TMB for treatment. Case No. 24 showed a mutation in the MSH6 gene. The TMB was 9, which is not in the TMB-High category by definition, but together with the MSH6 mutation, it was considered to be a target for treatment. Therefore, we have modified the expression TMB-High in Table 3, No.19 and No.24.

Reviewer #2

 Overall, this study’s goal is to describe the utility of Clinical genomic sequencing in Japan, a country that historically puts significant limitations on who can receive NGS sequencing. As a consequence, limited data is available from this country regarding NGS clinical utility.

I think the data they have collected could be useful to have in the literature, but as of now, the authors present the data purely in an observational format without providing any context or statistical comparisons about how their data in Japanese populations compare to previously published Foundation CDx retrospective studies.

Given the only novelty of this paper would be how numbers of clinically relevant alterations compare in populations in Japan vs other populations, statistical comparisons between this study's population and others are important to include as part of the results in order to provide context.

Reply: Thank you for your suggestion. We tried statistical comparison to foreign literatures. But it was difficult. We added this in limitation. 

I’m unsure about the use of the word “Actionability” in the context of their document. I believe the right phrase may instead be “Clinical Utility”. In my thought process, “actionability” would imply an alteration that would change the treatment options available to a patient. If possible, It would be helpful to report which of the alterations reported in Foundation Dx fall under “actionable”.

Based on the frequency reported (including an average of 3 “actionable” alterations/patient) I have to assume many of these alterations are only clinically significant in their prognostic abilities, not in their ability to open therapeutic options. Examples of these are TP53 or KRAS, which are clinically useful from a prognostic standpoint, but I feel calling them “actionable” may be disingenuous. However, I do not feel strongly about this semantic distinction if it doesn't bother the editor or other reviewers. 

Reply: Thank you for your comments. We explain the definition of actionable in discussion. 

From a recommendation standpoint, if they are able to provide statistical context for how the results in their population compare with previous Foundation CDx reports in USA and Europe, including alteration prevalence and power to detect statistical differences, I think the paper is worth adding to the literature via PLOSOne publication. (more detailed comments in the attached document).

Reply: Thank you for your suggestion. We tried statistical comparison to foreign literatures. But it was difficult. We added it in limitation. 

We also thank the review #2 for suggested changes in the manuscripts. We included these suggestions in the revised manuscript.

---

## [Decision Letter · Decision Letter 1]

15 Mar 2022

Clinical utility of Comprehensive Genomic Profiling in Japan: Result of PROFILE-F study.

PONE-D-21-35160R1

Dear Dr. Aoyagi,

We’re pleased to inform you that your manuscript has been judged scientifically suitable for publication and will be formally accepted for publication once it meets all outstanding technical requirements.

Kind regards,

Hyunseok Kang, MD, MPH

Academic Editor

PLOS ONE

Additional Editor Comments (optional):

Reviewers' comments:

Reviewer's Responses to Questions

**Comments to the Author**

1. If the authors have adequately addressed your comments raised in a previous round of review and you feel that this manuscript is now acceptable for publication, you may indicate that here to bypass the “Comments to the Author” section, enter your conflict of interest statement in the “Confidential to Editor” section, and submit your "Accept" recommendation.

Reviewer #1: All comments have been addressed

Reviewer #2: All comments have been addressed

2. Is the manuscript technically sound, and do the data support the conclusions?

Reviewer #1: Partly

Reviewer #2: Yes

3. Has the statistical analysis been performed appropriately and rigorously? 

Reviewer #1: N/A

Reviewer #2: Yes

4. Have the authors made all data underlying the findings in their manuscript fully available?

Reviewer #1: Yes

Reviewer #2: Yes

5. Is the manuscript presented in an intelligible fashion and written in standard English?

Reviewer #1: Yes

Reviewer #2: Yes

6. Review Comments to the Author

Reviewer #1: The authors reported the clinical utility of CGP test in Japan using commercial targeted sequencing system, and they concluded it might be useful for detecting gene alterations in various cancer types and exploring treatment options. The authors replayed to all of my comment.

Reviewer #2: I appreciate authors detailed in addressing concerns I brought up and for the diligence in adding the context of previous NGS clinical sequencing reports from other populations in how they compare to this particular study.

7. PLOS authors have the option to publish the peer review history of their article (what does this mean?). If published, this will include your full peer review and any attached files.

Reviewer #1: **Yes: **Hideyuki Hayashi

Reviewer #2: No

---

## [Editor Report · Acceptance letter]

23 Mar 2022

PONE-D-21-35160R1 

Clinical utility of Comprehensive Genomic Profiling in Japan: Result of PROFILE-F study. 

Dear Dr. Aoyagi:

I'm pleased to inform you that your manuscript has been deemed suitable for publication in PLOS ONE. Congratulations! Your manuscript is now with our production department. 

Kind regards, 

on behalf of

Dr. Hyunseok Kang 

Academic Editor

PLOS ONE